# Exploring Safety Culture in the ICU of a Large Acute Teaching Hospital through Triangulating Different Data Sources

**DOI:** 10.3390/healthcare11233095

**Published:** 2023-12-04

**Authors:** Ellen Liston, Enda O’Connor, Marie E. Ward

**Affiliations:** 1St James’s Hospital, D08 NHY1 Dublin, Ireland; oconnoen@tcd.ie (E.O.); maward@stjames.ie (M.E.W.); 2Department of Surgical Affairs, Royal College of Surgeons, D02 YN77 Dublin, Ireland; 3School of Medicine, Faculty of Health Sciences, Trinity College, The University of Dublin, D02 PN40 Dublin, Ireland; 4Centre for Innovative Human Systems, Trinity College, The University of Dublin, D02 PN40 Dublin, Ireland

**Keywords:** patient safety culture, psychological safety, just culture, intensive care, acute hospital

## Abstract

Safety Culture (SC) has become a key priority for safety improvement in healthcare. Studies have identified links between positive SC and improved patient outcomes. Mixed-method measurements of SC are needed to account for diverse social, cultural, and subcultural contexts within different healthcare settings. The aim of the study was to triangulate data on SC from three sources in an Intensive Care Unit (ICU) in a large acute teaching hospital. A mixed-methods approach was used, including analysing the Hospital Survey for Patient Safety Culture results, retrospective chart reviews using the Global Trigger Tool (GTT) for the ICU, and staff reporting of adverse events (AE). There was a 47% (101/216) response rate for the survey. Further, 98% of respondents stated a positive patient safety rating. The GTT identified 16 AEs and 11 AEs that were reported in the same timeframe. The triangulation of the data demonstrates the complexity of understanding components of SC in particular: learning, reporting, and just culture.

## 1. Introduction

Patient Safety has been recognised as an important public health, ethical, and economic issue needing significant research and improvement initiatives since the publication of landmark reports [1]. High-profile incidents and reports in healthcare have revealed multiple systemic failures attributable to Safety Culture (SC), including poor leadership, failure to risk assess, and breakdown of teamwork [2]. SC has become a key priority for safety improvement in healthcare, despite variations in definition [3]. The WHO Global Patient Safety Action Plan emphasises the need to instil a strong safety culture into the design and delivery of healthcare to ensure a reduction in both patient and staff harm [4]. Originating in the Nuclear Power industry, the concept of SC was coined to capture a lack of individual and organisational commitment to prioritise safety [5]. Research in the largest EU-funded project on Human Factors in aviation safety, the human integration in the lifecycle of aviation systems (HILAS) programme, adapted Reason’s initial five-component model of safety culture [6], to include the following seven components: (i) prioritising safety; (ii) ensuring standards and reliability; (iii) flexibility and resilience; (iv) learning culture; (v) teamwork; (vi) a reporting culture; and (vii) a just culture [7]. All these components, taken together, are important to SC [8].

Despite SC being a recognised priority, rates of patient harm have remained, if not increased, and patient safety initiatives have not systematically tackled SC within healthcare [2,9]. The diversity in healthcare organisations and systems accounts somewhat for this, but Provonost et al. argue that prioritising safety is key and taking a systems approach to do so rather than a persistent blame culture, focusing on punishment rather than learning, which appears to be characteristic of healthcare [10]. Under-reporting of incidents remains a widespread issue [11,12], and barriers to reporting identified in the literature include fear of repercussions and a lack of feedback or action when incidents are reported [13,14,15,16]. Mixed-method measurements of SC are needed to account for diverse social, cultural, and subcultural contexts within different healthcare settings [17,18] and to understand how these different components of SC play out in different settings. To date, there is no study in Ireland that has used more than one method of measurement to examine SC, specifically in the ICU.

### 1.1. Safety Culture in the ICU

Within hospitals, intensive care units (ICUs) are particularly high-risk areas for medical errors and adverse events (AE) that could occur due to the complexity of care and the patients’ fragile medical conditions [18]. Studies have been carried out using survey and interview-based designs to explore SC in the ICU. Farzi et al. [19], for example, studied safety culture among Iranian nurses in the ICU using the Agency for Healthcare Research and Quality’s (AHRQ) Hospital Survey on Patient Safety Culture (HSOPSC) and reported that nurses found “teamwork within units” and “organizational learning-continuous improvement” to be the most positive features of their safety culture. They assigned the least positive scores to “handoffs and transitions” and “non-punitive response to errors”. Gomides et al. [20] evaluated perceptions of SC within an MDT of a Brazilian ICU using the Safety Attitudes Questionnaire. They found low scores for safety attitudes related to the domains of management, working conditions, and communication failures.

Tlili et al. [18] studied SC among nurses, doctors, healthcare technicians, and assistant caregivers in 15 Tunisian ICUs using the HSOPSC and follow-up interviews with 12 staff. They found all the SC dimensions had a score of less than 50%. The most widely implemented dimension was “teamwork within units”. The least positive scores were assigned to “communication openness” and “non-punitive response to error”.

### 1.2. Evaluating SC and Triangulating Data

#### 1.2.1. SC Survey Studies

There are a limited number of studies that use more than one method to evaluate SC. Recent studies [17,21,22,23,24] have emphasised the importance of using various data collection methods to account for the complexity of SC. The most frequently used measurement tools in the SC literature are the Safety Attitudes Questionnaire and the AHRQ HSOPSC [1] surveys. Both questionnaires have been empirically tested for psychometric validity and reliability [16,25]. HSOPSC has been recommended for use in healthcare organisations to assess SC, track changes, and evaluate the impact of safety interventions [11]. The AHRQ also allows organisations to enter their results into a database that allows for the comparison of findings across similar contexts.

#### 1.2.2. SC Mixed-Methods Studies

Mardon et al. [21] measured staff perceptions of SC and their association with AE using the HSOPSC and AHRQ Patient Safety Indicators. They found that higher SC scores tended to have fewer Patient Safety Indicators, including, for example, lower rates of in-hospital complications, and that this association remained after adjustments for hospital characteristics and measurements. More recently, Denning et al. [23] also investigated SC by studying staff perceptions of SC and staff AE reporting, specifically in response to the COVID-19 pandemic, and found there was a considerable reduction in reporting. When broken down by type, “No Harm” and “Near Miss” reports were significantly reduced during the pandemic (*p* < 0.003). However, the level of reporting for “Harm” incidents did not significantly change.

#### 1.2.3. Adverse Event Reporting and Medical Record Review Studies

A review by Hibbert et al. [26] examined the available literature on two-stage Medical Record Reviews, as an adjunct to reported AE [26]. The two most commonly used Medical Record Reviews in the literature are the Institute for Healthcare Improvement Global Trigger Tool (GTT) and the Harvard Method. They found that GTT is a more commonly used tool, as it is a flexible tool that can be modified to the context. Rates of AE per 100 admissions ranged from 7% to 51%, with 10% of admissions being associated with an AE [15].

The aim of this study was to explore SC in the ICU of a large acute teaching hospital through the rigorous application of three different methods and triangulating data from three sources:(i)Staff completion of the HSOPSC survey;(ii)Using the GTT to carry out a retrospective chart review of a sample of patient charts;(iii)Reviewing AEs as reported by staff through the hospital electronic AE reporting system.

## 2. Materials and Methods

### 2.1. Study Setting

The study took place in the largest acute teaching hospital in Ireland, with over 1000 beds and 5000 staff. The hospital provides acute and emergency care for adult patients through a broad range of specialist medical, surgical, and oncology services. The study was completed in the 26-bed ICU of the hospital. All 216 professionals who worked in the ICU during the time of the study were invited to participate. This included nurses (staff nurses, clinical nurse managers, clinical nurse facilitators, and advanced nurse practitioners), doctors (interns, senior house officers, registrars, and consultants), health care assistants, and health and social care professionals (HSCPs) (e.g., pharmacists, speech and language therapists, clinical nutritionists, medical social workers, occupational therapists, and physiotherapists). Based on the AHRQ guidance [26], for settings with 500 or fewer staff, a 50% response rate should be expected. Therefore, the predicted response rate was 50% (108) participants.

### 2.2. Study Design

A mixed-methods approach was used, building on the strengths of different approaches. Information on the SC in the ICU was explored through three sources:(i)Staff completion of the HSOPSC survey: Survey data was collected using HSOPSC version 2.0 (Appendix A) approved by the AHRQ [27]. The survey contains a total of 40 survey items, divided across 10 composite measures of SC, primarily using 5-point agreement scales (“Strongly disagree” to “Strongly agree”) or frequency scales (“Never” to “Always”) with a “Does not apply or Don’t know” option. Composites include Communication About Error, Communication Openness, Handoffs and Information Exchange, Hospital Management Support for Patient Safety, Organizational Learning-Continuous Improvement, Reporting Patient Safety Events, Response to Error, Staffing and Work Pace, Supervisor, Manager, or Clinical Leader Support for Patient Safety and Teamwork. There is also an item on patient safety events the respondent has reported and a further item seeking an overall rating on patient safety for their work unit/area. An additional open-ended question probes perceptions around overall patient safety in the hospital. The demographic data of participants were collected using the same questionnaire, identifying profession, work area, and level of experience. The wording of job roles was adapted to correlate with the Irish setting, as advised by the AHRQ guidance document.(ii)Using the GTT to carry out a retrospective chart review of a sample of patient charts: As many studies have previously identified low rates of incident reporting [9], a retrospective chart review was completed for a two-week period. The GTT for intensive care was used (Appendix A) to gather this data [28,29,30].(iii)Reviewing AEs reported by staff from the ICU through the hospital electronic incident reporting system.

### 2.3. Data Collection Tools, Methods, and Procedures

How data was gathered using the survey, GTT, and AE analysis is outlined here:(i)The HSOPSC, with a participant information leaflet (PIL), was distributed both electronically by email and with paper hard copies. Email links were sent by the critical care administrator, clinical nurse facilitator, and critical care lead to all staff working within the ICU at the time of the study. As the survey was completed anonymously, the Ethics Committee agreed that informed consent was implied through the completion of the survey. An additional follow-up email was sent one month later. To protect against unauthorised access, the survey link was distributed using the hospital internal email system. Hardcopies of the survey and PIL were distributed in person to all professionals working within the ICU at ward rounds, MDT meetings, and quality and audit meetings during February and March 2022.(ii)Prior to administration of the HSOPSC samples of patients’ electronic patient records (EPR) were reviewed using the GTT for predicting potential patient harm, as outlined in the Institute for Healthcare Improvement protocol [29,30]. Ten records were selected at random from all patients admitted and discharged from ICU, over a two-week period in February 2022. Patients who were admitted <24 h to ICU, under 18, and those patients with a primary psychiatric diagnosis were excluded as outlined in the Institute for Healthcare Improvement guideline. A random number generator was used to ensure random selection. Selected patient charts were anonymised on a password-protected Excel file, stored in the Principal Investigator’s secure folder on the hospital’s internal IT system. EPR records were reviewed for triggers initially by two reviewers, including the main author (EL) and an HSCP colleague who was familiar with the tool. Both reviewers had completed training in the use of the tool. Data was recorded on separate paper forms by each reviewer, as outlined in the Institute for Healthcare Improvement guide [30].(iii)Reviewing AEs was carried out through an analysis of AEs reported, through the hospital electronic incident reporting system, by staff who work in the ICU. All incidents reported in February 2022 were included. Data provided to the researchers was anonymised.

The Checklist for Reporting Of Survey Studies (CROSS) quality appraisal tool for carrying out web-based and non-web-based surveys was used and can be found in the Appendix A to this article [31].

### 2.4. Data Analysis

How the data gathered from the three different sources was analysed is outlined here:(i)Electronic and hard copy HSOPSC survey data were inputted into a Microsoft Excel spreadsheet created and supplied by the AHRQ. All included data were cross-checked for any errors in data entry before analysis. Responses were calculated referencing the AHRQ Guidance document [27]. Missing data and “Does not apply/Don’t know” responses were excluded from calculations. Descriptive statistics were used to calculate the mean score and the average percentage of positive responses. The average percentage of positive and negative responses was calculated. The scores were reversed for negatively worded items—these are noted with an (R) after them in the survey. A One-way ANOVA was used for comparison with the AHRQ international database, with statistical significance set at *p* < 0.05. Tukey post-hoc tests were used when significance was detected. Demographic data were analysed using descriptive statistics, referencing characteristics of respondents, specifically professional experience. Qualitative data from the open-ended questions of the questionnaire were analysed using content analysis [32]. The data was systematically reviewed and highlighted where SC domains or components were referenced. This was then coded and categorised by SC domain relevance. Further examination of the data was then completed to include barriers and facilitators to patient safety and SC, coded again by SC domains. Categorisation was then completed by grouping the data by positive or negative responses. Following this systematic analysis of the data, themes were formed.(ii)For the GTT, the triggers were examined for AE occurrence and classified in terms of harm. The National Coordination Council for Medication Error Reporting and Prevention (NCC MERP) harm categories are commonly used with the GTT [26] and include Category E: Temporary harm to the patient and required intervention; Category F: Temporary harm to the patient and required initial or prolonged Hospitalisation; Category G: Permanent patient harm Category H: Intervention required to sustain life; Category I: Patient death. After the initial review was completed separately by the two reviewers, the data were collated, and agreement was reached through consensus on rating. An ICU Consultant (EOC) then reviewed the collated data to ensure agreement on the occurrence of an AE and categorisation.(iii)The reported AE data from the hospital electronic incident reporting systems were reviewed and incidents reported were categorised by one author (EL) according to type, level of harm, and whether the reported AE was related to patients or staff. There was no other formal method of AE reporting in the hospital at the time of the study.

## 3. Results

The results will initially be presented separately and then triangulated to explore the SC in the ICU from the perspective that the different data sources give us.

### 3.1. Hospital Survey on Patient Safety Culture Results (HSOPSC) Results

#### 3.1.1. Demographics

A total of 101/216 questionnaires were completed from February 2022 to the end of March 2022. This equated to a 47% response rate. Table 1 shows a further breakdown of respondents by staff position. Further, 97% of respondents worked only in Intensive Care, with 2% in anaesthesiology and 1% across many different units. Staff experience varied, with 17 respondents working in the hospital less than a year, 41 respondents 1–5 years, 19 respondents 6–10 years, and 24 respondents 11 years or more. 79 respondents worked 30 to 40 h per week, with 21 working more than this. A total of 96 respondents (95%) reported having direct contact with patients.

#### 3.1.2. Survey Composites

Table 2 shows the percentages of positive, neutral, and negative responses from staff for SC composite survey items. Where the question was positively worded, answers of “Strongly Agree/Agree” and “Always/Most of the time” account for positive responses. In negatively worded questions that were reverse scored (R), answers of “Strongly Disagree/Disagree” and “Never/Rarely” account for positive responses. Negative answers were the opposite of this. Neutral responses account for answers of “Neither Agree nor Disagree/Sometimes”. Where a respondent answered “Does not know” or “Does not Apply” their answer was not included in the percentage.

Figure 1 shows the average positive response of respondents for each of the 10 SC composites contained in the survey, in comparison to the AHRQ database (172 entries in the database as of December 2021). The Kolmogorov-Smirnov statistical test was performed on both data sets and found the data to be normally distributed. One-way ANOVA tests were completed using IBM^®^ SPSS^®^ Version 26. Results showed differences were not statistically significant (*p* = 0.07–0.277), and post-hoc Tukey tests were therefore not performed.

In the Teamwork composite, there was an overall 90% positive response, with the included individual question responses in the MAX category. Composites of communication openness, reporting patient safety events, communication about errors, and hospital management support for patient safety were at least 10% below the average. When compared to data for the ICU, this only applied to the composites of communication openness and reporting patient safety events.

Individual questions in Communication openness scored 6% below the minimal recorded on the database and 23% below the average for “When staff in this unit see someone with more authority doing something unsafe for patients, they speak up”. For “In this unit, staff are afraid to ask questions when something doesn’t seem right” the responses were 16% below average. These relate to the SC component of just culture. In “Reporting Patient Safety Events” the individual question “When a mistake is caught and corrected before reaching the patient, how often is this reported” scored 20% below the average (45%)—this is related to the SC component of reporting culture and in particular reporting of “near misses”.

Communication about Error scored 19% below the average for “We are informed about errors that happen in this unit”. In “Hospital Management Support for Patient Safety”, the individual statement “Hospital management seems interested in patient safety only after an adverse event happens” scored 24% below the average positive response, with a 58% negative response.

#### 3.1.3. Safety Rating

A total of 98% of respondents reported at least a good safety rating, of whom 48% reported a very good rating, 15% excellent, and 35% good. These were compared favourably to AHRQ data (91% of respondents reported positively, 7% reported fair, and 2% reported poor). When adjusted to ICU specifically, there was a slight increase to 10% of fair responses and 3% of poor responses; however, again, the most common rating was very good (41%).

#### 3.1.4. Number of Events Reported

Here, 48% of respondents had not completed an incident report in the preceding 12 months, 37% had completed 1–2 reports, and 14% had completed more than two. This was comparable to the AHRQ database, with a slight increase in those reporting more than two when adjusted to the ICU specifically (26%).

#### 3.1.5. Qualitative Data

Here, 18% (n = 18) of respondents completed the open-ended question (Table 3). Content analysis was carried out on the responses, identifying both positive and negative themes and categorising them based on the safety culture domains where relevant.

Of the responses to the open-ended qualitative question, there was one positive comment relating to teamwork; the other 17 comments were either negative or suggestions for change. The themes for these related to staffing concerns (n = 7), communication (n = 5), reporting culture (n = 3), equipment (n = 2), and others (2).

### 3.2. AE Results

In total, there were 11 reported AEs related to the ICU for February 2022 (Table 4). Of these, six related to pressure ulcers, two to medication safety, two to device/equipment faults, and one to a reported occupational health injury. Further, 10 (90%) of these events affected patients, with the remaining incident affecting a staff member. The AEs ranged in severity from near miss (n = 1) to harm events (n = 10). The reported pressure ulcers were of low grade and acquired during the patient’s ICU stay.

In the hospital, medication-related harm is separated from other types of harm and categorised using the NCC MERP index for categorising medication errors. Of the two medication-safety AEs, one was classified as a MERP category “C” (An error occurred that reached the patient but did not cause patient harm) and one as a MERP category “E” (An error occurred that may have contributed to or resulted in temporary harm to the patient and required intervention).

### 3.3. GTT Results

Of the 10 charts reviewed using the GTT, a total of 43 triggers were identified, with a median number of 4 per chart (range 2–7). From this, 16 AEs were identified (Table 4); the median was 2 per chart (range 0–4), and 2 charts (20%) contained no adverse events. 15 of the 16 AEs were rated as NCC MERP category E (temporary harm to the patient and required intervention) and 1 category F (temporary harm to the patient and required readmission to the ICU or prolonged hospitalisation either in the ICU or step-down units). Six of the AEs were related to medication, including antibiotics, anticoagulants/antiplatelets, insulin, and narcotics. The severity of harm from these events was: 1 MERP category “C” (An error occurred that reached the patient but did not cause patient harm); 3 MERP category “D” (An error occurred that reached the patient and required monitoring to confirm that it resulted in no harm to the patient and/or required intervention to preclude harm); 2 MERP category “E” (An error occurred that may have contributed to or resulted in temporary harm to the patient and required intervention). The median length of stay in the ICU in the charts included was 6.5 days (range 1–14).

### 3.4. Triangulation of Data

#### 3.4.1. HSOPSC and AE Data

The triangulation of data relates most to the SC components of learning, reporting, and just culture. When the composites from HSOPSC were explored in more detail, it could be seen that ‘reporting patient safety events’ is at 58%, 16% below the average AHRQ score. Participants were also asked about their own AE reporting behaviour in the survey. 48% of respondents reported that they had not completed an incident report in the last 12 months, with 37% of respondents stating they had completed 1–2 and 14% of respondents stating they had completed more than two. Taken together this roughly works out for staff noting that they report about one AE every 4 days. The actual AE reported data for a two-week timeframe was 11 AEs reported—this corresponds to nearly one AE every day. So staff reported more AEs to the hospital AE reporting system than they noted in the HSOPSC that they do. However, reporting may be below the national average. Reported AEs at the same time point showed a 9% occurrence, 3% lower in comparison to large-scale published Irish studies [14]. While not directly comparable, due to the small data set, it may demonstrate an underreporting of AEs within the study setting.

#### 3.4.2. GTT and AE Data

When we compare the GTT results with the reported AEs, we find that 16 AEs were identified in the GTT, none of which were reported as AEs through the hospital incident reporting system. None of the reported AEs (11) through the hospital system were the same events as the AEs identified using the GTT (16). This might help explain the perception of staff that they underreport AEs. Staff may realise that more AEs are taking place than they report, and this may be related to the nature of the AEs (e.g., near misses). In February 2022, there were 123 admissions to the General Intensive Care Unit. The GTT data therefore represents 8% of these admissions. Within the 8% represented, an AE occurred in 80% (n = 8) of charts reviewed using the GTT. Further, 94% of the AEs identified through the GTT caused temporary or no harm to the patient, which may account for some of these events not being reported. For example, specifically focusing on medication-related events, there were two events reported in February 2022, whereas the GTT data identified six medication safety events within the data set, none of which were the same as the reported events. As stated above, four of these events did not cause harm to the patient, with the other two causing temporary harm, which may account for them not being reported.

## 4. Discussion

The survey data demonstrate an overall positive SC, with 5 out of the 10 composites scoring the same as, above, or well above the international benchmark. Areas of Teamwork, Supervisor/Manager/Clinical leader support for patient safety; Organisational learning/Continuous improvement; and Handoffs information exchange scored above 75% positive responses, at the upper range when benchmarked against the AHRQ database. In the 2022 report of OECD countries, including over 20 countries, 68% of health workers reported high levels of teamwork, and 65% stated their organisation exhibited positive continuous improvement [1]. For the study population, there was a 22% higher positive response for Teamwork and 12% for Continuous Improvement. This demonstrates a willingness to support and work effectively together towards improved patient outcomes, evaluate and implement change, and take action to prevent AEs [27]. The SC dimensions of teamwork, ensuring standards and reliability, and a learning culture correlate with these composites.

Areas for improvement identified in the survey results included communication openness, hospital management support for patient safety, staffing and work pace, communication about errors, and reporting patient safety events. While all achieved a positive score of over 50%, all 5 composites were below the international benchmark. Similar areas of improvement were found in previous studies, specifically, staffing and work pace, communication about errors, and reporting patient safety events. In addition, handoffs and transitions also needed improvement [21,22]. When we triangulate these results, with those from the GTT and AE, the overall results relate most to the three SC components of prioritising safety, reporting, and just culture.

### 4.1. Prioritising Safety

There is a significant difference between perceived Supervisor/Manager support for patient safety and Hospital management support despite both composites correlating with the Prioritising Safety component. This dichotomy may be explained by staff perceiving their local managers to be more involved in developing a good safety culture while perceiving managers more removed from them to be less involved in the unit’s safety culture. While the hospital has an active quality and safety improvement directorate supporting over 20 programmes of work addressing key risks within the hospital, front-line staff may not be as intimately aware of these and other patient safety efforts managed and delivered by colleagues less closely connected with them in the hospital. These findings are similar to those from other national studies [33,34,35]. This is also highlighted by the WHO Global Patient Safety Action Plan and the Health Service Executive in Ireland Patient Safety Strategy emphasising the need to build leadership and management capacity at all levels of the health system to ultimately achieve more resilient systems [4,36].

### 4.2. Just Culture

The low positive score in certain composites of the HSOPSC may indicate a lack of psychological safety and trust of staff to be supported by management, prerequisites to a just culture. Psychological safety relates to how individuals perceive their work environment as being supportive of them being able to ask questions and express concerns [37]. A just culture refers to “an atmosphere of trust in which people are encouraged, even rewarded, for providing essential safety-related information—but in which they are also clear about where the line must be drawn between acceptable and unacceptable behaviour” [6].

When there is a lack of psychological safety staff are less likely to speak up or report errors in fear of blame, resulting in a widening of the gap between work-as-done by clinicians and work-as-imagined by managers, resulting in less resilient systems of care [38]. This has a significant impact on patient safety, preventing quality improvement as underlying problems are not identified, resulting in failures to prevent the recurrence of AEs [38]. This is a key area for system improvement that can be addressed by acknowledging fallibility, actively seeking and valuing staff input, and addressing any hierarchical concerns of staff [37,38]. This has been highlighted as a key focus nationally in Ireland, requiring a multifaceted approach involving all stakeholders, where behaviours that remove fear of reprisal if incidents are reported and build trust that errors are managed fairly are embedded from the top down [39].

### 4.3. Reporting Culture

These results suggest a reluctance to report harm and poor communication of errors [27]. This was further emphasised in the qualitative data, with themes arising around communication issues and poor reporting culture. The association between lower SC scores and lower incident reporting was also seen in previous studies using more than one method of analysis [21,22]. When it comes to AE reporting, some reduction in reporting levels may be attributable to the impact of the COVID-19 pandemic at the time. The ongoing COVID-19 pandemic has caused increased emotional distress, burnout, and staff turnover in healthcare, especially in ICUs [40,41]. This study was completed after several waves of COVID-19, and burnout and exhaustion experienced by staff may have contributed to reduced reporting and impacted the SC. Internationally, studies have shown there has been a reduction in reporting during COVID-19, estimated at 4.4% overall [23,42,43]. When examined in terms of category, the reduction was in “no harm” and “near miss” AEs, with little change in the number of “harm” events reported, which seems to be reflected in our study [23]. Various factors may have influenced this, such as increased workload, lower staffing levels, and time to report, and changed the perception of the importance of AE reporting, especially near misses and no-harm events [43]. The reduction in reporting may represent potential lost learning opportunities to improve patient safety during an unprecedented time [23]. The hospital’s safety and risk management policies strongly encourage AE reporting in the case of harm or near miss, related or unrelated to the provision of care, regardless of the severity.

## 5. Conclusions

All seven components of SC need to be present to achieve a positive SC and have a lasting impact on patient safety [6,7]. This study demonstrates the importance of examining all the components of SC from different perspectives. While our triangulation sheds the most light on the components of prioritising safety, reporting, and just culture, it has shown that, when looked at from different perspectives, improvements are needed in all these areas. Some authors would argue that psychological safety, speaking up about safety, and a supporting just culture, which ensures we are not blamed for mistakes (but held to account when we do deliberate harm), are the most important building blocks of a positive SC (e.g., [37,38]).

Changing culture, and SC in particular, is an essential component of whole system change [44]. There are few case studies or reports of successful system change in healthcare, and there is a lack of agreement on “whole system change” [45]. To achieve improvements in patient safety, there first needs to be a shift towards a whole systems approach with a clear understanding of the goals of that system—patient safety must be a key goal of our healthcare system and perceived by all staff as a key priority for the organisation, as outlined in the WHO Global Patient Safety Action Plan [4,45]. A whole systems approach prioritising patient safety would then enable collaborative working to achieve sustained improvements in patient safety. Successful large-scale projects aiming to reduce harm in the ICU have been founded on creating a positive SC that meaningfully engages clinicians, patients, and families [46,47,48] and have adopted methodologies focused on strong system designs with goals of safety, high efficiency, and value [48]. The findings from this study have been presented to the ICU MDT and at fora across the hospital. A SC improvement plan has been put in place and learning from this study has contributed to ongoing system change for quality and safety improvement in the hospital.

### Limitations and Future Research

There are some limitations to the study. Understanding SC is very difficult. To really achieve triangulation on the different components of SC as per the adaptation of the Reason model [7,8] would have required a much larger study. The triangulation of data in this study was achieved for three components of safety culture, which are priority, reporting, and just culture. As outlined by the WHO [4], the engagement of patients and families in the assessment of SC would likely have provided another aspect of assessment and should be considered for future, larger-scale studies. This is a one-time-point study with all data gathered within a four-week timeframe. It would have been beneficial to complete further chart reviews over a longer time period, although the charts reviewed were randomly selected as per Institute for Healthcare Improvement guidance, and reviewing additional charts may not have added anything of further significance to the study. As this study was completed after multiple COVID-19 waves, emotional distress, staff burnout, staff turnover, and changes in the work environment may have significantly impacted the findings from the study. This may have influenced the low response rate, despite attempts at recruiting more participants through staff meetings and email reminders, as outlined above.

## Figures and Tables

**Figure 1 healthcare-11-03095-f001:**
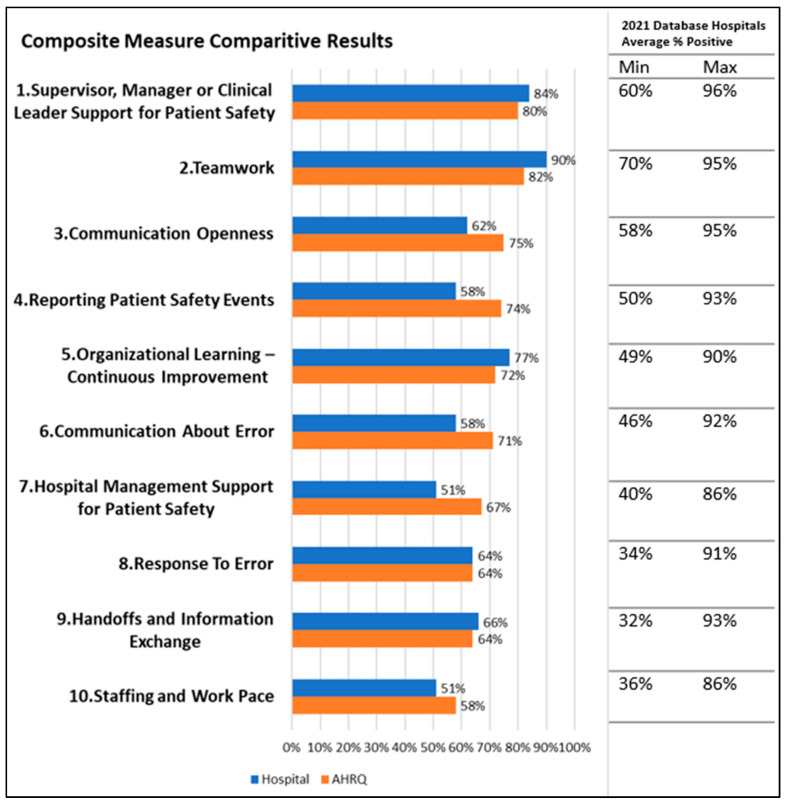
AHRQ database vs. study results composite comparison.

**Table 1 healthcare-11-03095-t001:** Breakdown of respondents by staff position.

Staff Positions	Percentage
Advanced Practice Nurse (ANP, CNS, CF, CNM)	17%
Staff Nurse	56%
Intern/SHO Doctor	4%
Registrar or Consultant Doctor	10%
Health and Social Care Professional (HSCP)	11%
Other (incl. 1 non-disclosed)	3%

Advanced Nurse Practitioner(ANP), Clinical Nurse Specialist (CNS), Clinical Facilitator (CF), Clinical Nurse Manager (CNM).

**Table 2 healthcare-11-03095-t002:** Breakdown of HSOPSC responses per survey question.

		Positive %	Neutral %	Negative %
**Supervisor, Manager or Clinical Leader Support for Patient Safety**				
1. My supervisor, manager, or clinical leader seriously considers staff suggestions for improving patient safety.		84	12	4
2. My supervisor, manager, or clinical leader wants us to work faster during busy times, even if it means taking shortcuts. (R)		77	15	8
3. My supervisor, manager, or clinical leader takes action to address patient safety concerns that are brought to their attention.		93	6	1
**Teamwork**				
1. In this unit, we work together as an effective team.		97	2	1
2. During busy times, staff in this unit help each other.		97	3	0
3. There is a problem with disrespectful behaviour by those working in this unit. (R)		76	12	12
**Communication Openness**				
1. In this unit, staff speak up if they see something that may negatively affect patient care.		76	21	3
2. When staff in this unit see someone with more authority doing something unsafe for patients, they speak up.		49	29	22
3. When staff in this unit speak up, those with more authority are open to their patient safety concerns.		66	30	4
4. In this unit, staff are afraid to ask questions when something does not seem right. (R)		55	38	7
**Reporting Patient Safety Events**				
1. When a mistake is caught and corrected before reaching the patient, how often is this reported?		45	28	28
2. When a mistake reaches the patient and could have harmed the patient but did not, how often is this reported?		71	18	10
**Organizational Learning—Continuous Improvement**				
1. This unit regularly reviews work processes to determine if changes are needed to improve patient safety.		78	14	8
2. In this unit, changes to improve patient safety are evaluated to see how well they worked.		75	16	9
3. This unit lets the same patient safety problems keep happening (R)		80	9	12
**Communication About Error**				
1. We are informed about errors that happen in this unit.		51	39	10
2. When errors happen in this unit, we discuss ways to prevent them from happening again.		68	25	7
3. In this unit, we are informed about changes that are made based on event reports.		56	31	13
**Hospital Management Support for Patient Safety**				
1. The actions of hospital management show that patient safety is a top priority.		66	22	12
2. Hospital management provides adequate resources to improve patient safety.		63	15	23
3. Hospital management seems interested in patient safety only after an adverse event happens. (R)		25	17	58
**Response To Error**				
1. In this unit, staff feel like their mistakes are held against them. (R)		61	22	18
2. When an event is reported in this unit, it feels like the person is being written up, not the problem. (R)		62	14	24
3. When staff make errors, this unit focuses on learning rather than blaming individuals.		66	20	14
4. In this unit, there is a lack of support for staff involved in patient safety errors. (R)		67	21	12
**Handoffs and Information Exchange**				
1. When transferring patients from one unit to another, important information is often left out. (R)		47	16	37
2. During shift changes, important patient care information is often left out. (R)		74	12	15
3. During shift changes, there is adequate time to exchange all key patient care information.		77	7	16
**Staffing and Work Pace**				
1. In this unit, we have enough staff to handle the workload.		50	11	40
2. Staff in this unit work longer hours than is best for patient care. (R)		32	23	44
3. This unit relies too much on temporary, float, or PRN staff. (R)		60	25	15
4. The work pace in this unit is so rushed that it negatively affects patient safety. (R)		62	19	19
	**Positively worded**	**Negatively Worded (R)**		
**Positive**	“Strongly Agree/Agree” “Always/Most of the time”	“Strongly Disagree/Disagree” “Never/Rarely”		
**Neutral**	“Neither Agree nor Disagree/Sometimes”.	“Neither Agree nor Disagree/Sometimes”.		
**Negative**	“Strongly Disagree/Disagree” “Never/Rarely”	“Strongly Agree/Agree” “Always/Most of the time”		

**Table 3 healthcare-11-03095-t003:** Sample qualitative comments.

Sample Qualitative Comments		Staff Category
Teamwork positive comment	“Fantastic teamwork between all forms of staff.”	Other/Not disclosed
Staffing Concerns	“Adequate staffing would positively impact patient safety, ensuring they are receiving the correct level of care”.	Health & Social Care Professional (HSCP)
Communication	“I would also like to see a clear follow through and delivery of information to staff. How many near misses/adverse events etc. and what is being done to minimise further incidents”	Advanced Nurse Practitioner (ANP, CNS, CF, CNM) *
Reporting Culture	“I have been blamed for putting too many incident reports, so now avoid them as far as possible.”	Staff Nurse
Equipment	“Sometimes lack of equipment, bed spaces too small/not fit for purpose can cause safety concerns, which are well known, but little has changed.”	Staff Nurse

* Advanced Nurse Practitioner(ANP), Clinical Nurse Specialist (CNS), Clinical Facilitator (CF), Clinical Nurse Manager (CNM).

**Table 4 healthcare-11-03095-t004:** Breakdown of AEs.

	AE Data	GTT
ICU admissions	123	123
Charts reviewed	-	10
Triggers	-	43
No. of events	11	16
Near miss	1	0
Temp harm	3	15
Harm	7	1
Medication-related	2	6

## Data Availability

The data presented in this study are available on request from the corresponding author. The data are not publicly available due to the sensitivity of the topic.

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
