# Peer review of "Exploring Safety Culture in the ICU of a Large Acute Teaching Hospital through Triangulating Different Data Sources"

_healthcare, 2023, doi:10.3390/healthcare11233095_

Round 1
Reviewer 1 Report
Comments and Suggestions for Authors
The manusript reports excellently a safety culture evaluation and check in a hospital. The value of the paper lays in showing how such an analysis can be made and what should be taken into account.
On the other side, the manuscript fails to build any theoretical framework that would base on earlier research, and would make it possible to make findings that can be generalized. Now the article remains a practical case study without clear scientific contribution.
Smaller issues:
- all acronyms used should be defined when used first time
- the structure of the titles is odd, with long paragraphs not belonging to no subtitle
- the article tells about analyzing of several data sets, but does not give details on how this analysis was done
Comments on the Quality of English LanguageSome minor errors exist
Author Response
Dear Reviewer One,
Many thanks for you feedback and suggestions. We have taken these on board and we hope addressed the errors that you have outlined.
- all acronyms used should be defined when used first time- Thank you for your comment we have since corrected this, an error that occurred when many drafts were made.
- the structure of the titles is odd, with long paragraphs not belonging to no subtitle- We have tried to clear this up and include further subtitles to provide more clarity.
- the article tells about analyzing of several data sets, but does not give details on how this analysis was done.
We have added more detail regarding the data analysis to try address this. The majority of the other analysis isn’t described as we followed the AHRQ guidance directly for the survey results which is free access and the IHI guidance for the trigger tool, therefore the detail of this can be accessed should a reader require it. I am more than happy to add further detail if you think it is important rather than referencing the guidance documents.
Furthermore we have included more reference to previous studies to make the results more generalisable. We hope the changes made have improved the study and address your feedback.
Kind regards,
Ellen Liston
Reviewer 2 Report
Comments and Suggestions for Authors
In general, study objectives were achieved. Never the less, here are some notes that caught my attention:
- regarding the few studies that use more than one method to assess CS (73), I ask if the results compare with those achieved by this study? Just to understand if the most problematic areas are identical (you could include it in the discussion?)
- in addition to AEs reported by staff through the hospital electronic AE reporting system (103), is there any other adverse event report system carried out by the ICU or pharmacists? This is to exclude other potential reports.
- (149) the term PIL should have been designed previously (Patient Information Leaflet)
- (219-220. Table 1) – Does the distribution presented matches with the distribution of ICU workers categories? Just to understand the representation
- pg. 7 organizational learning – continuous improvement. The results of question 3 contradicts the previous 2 questions and this is not addressed in the discussion
- (476) – “this may have” or “this may be” ???
Comments on the Quality of English Language- (476) – “this may have” or “this may be” ???
Author Response
Dear Reviewer Two,
Thank you for reviewing our paper and provided us with such helpful feedback. We hope we have addressed you questions with the updated manuscript.
In response to your suggestions:
In general, study objectives were achieved. Never the less, here are some notes that caught my attention:
- regarding the few studies that use more than one method to assess CS (73), I ask if the results compare with those achieved by this study? Just to understand if the most problematic areas are identical (you could include it in the discussion?)
Similar results were found while not identical, I have added reference to these now in the discussion.
- in addition to AEs reported by staff through the hospital electronic AE reporting system (103), is there any other adverse event report system carried out by the ICU or pharmacists? This is to exclude other potential reports.
There was no other formal method of AE reporting in the hospital at the time of the study.
- (149) the term PIL should have been designed previously (Patient Information Leaflet)
Thank you I have since corrected this.
- (219-220. Table 1) – Does the distribution presented matches with the distribution of ICU workers categories? Just to understand the representation
Great point, yes the response to the data does roughly represent the general staff distribution in the ICU. The breakdown of staffing is 70% nursing, 15% medical, 10% HSCPs and 5% support workers/healthcare assistants.
- pg. 7 organizational learning – continuous improvement. The results of question 3 contradicts the previous 2 questions and this is not addressed in the discussion.
As question 3 in this Safety culture domain is negatively worded the calculations were reversed, meaning the 80% positive result correlates with respondents not agreeing with the statement “This unit lets the same patient safety problems keep happening”
- (476) – “this may have” or “this may be” ??? I have rephrased this sentence- hopefully it now reads more easily.
We hope that this addresses your feedback and look forward to hearing from you.
Kind regards,
Ellen Liston
Reviewer 3 Report
Comments and Suggestions for Authors
Patient safety in ICU is very important topic nowadays with near miss error and adverse events.
1. Introduction- literature review of previous studies and the status of safety culture in ICU are suggested but the necessary of this study need to be suggested clarified.
The abbreviation is too many and is confused.
2. In methods, HSOPSC was surveyed using 5-Likert scale but in table the data were suggested with positive, neutral, negative results.
3. In Figure 1, hospital vs AHRQ were compared. In the data analysis, it was written that ANOVA was used but the analysis used in Figure 1 was Chi-test. The p-value are not suggested in Figure 1.
Which year of AHRQ data did the author use to compare with this collected data(hospital?)
4. Table 3 was the results of qualitative comments. What methods did the authors use to analysis the data?
To publish this manuscript, triangulation methods need to suggested with scientific evidence.
Author Response
Dear Reviewer 3,
Thank you for your detailed feedback and guidance. We hope we have addressed your below questions with the following:
1. Introduction- literature review of previous studies and the status of safety culture in ICU are suggested but the necessary of this study need to be suggested clarified- We have included additional detail to try make this more clear.
The abbreviation is too many and is confused. - We have reduced the abbreviations where possible to try improve the clarity
2. In methods, HSOPSC was surveyed using 5-Likert scale but in table the data were suggested with positive, neutral, negative results.
Yes the survey uses 5point likert scale. To display the difference between positive and negative results the two positive points were grouped and so were the two negative points on the scale.
3. In Figure 1, hospital vs AHRQ were compared. In the data analysis, it was written that ANOVA was used but the analysis used in Figure 1 was Chi-test. The p-value are not suggested in Figure 1.
Yes Figure 1 demonstrates the direct comparison of the hospital data Vs AHRQ database averages. One way anova was completed but no significance detected and therefore not displayed in figure 1 to try keep the figure easy to read.
Which year of AHRQ data did the author use to compare with this collected data(hospital?) AHRQ data for 2021 which includes 172 data sets was used, this is outlined in the results section description.
4. Table 3 was the results of qualitative comments. What methods did the authors use to analysis the data? Many thanks for highlighting the limited detail provided in the initial manuscript, we have now added a description of this analysis in the methods section.
We hope the updated manuscript has addressed your concerns and look forward to hearing from you.
Kind Regards,
Ellen Liston